# Transferrin Isoforms, Old but New Biomarkers in Hereditary Fructose Intolerance

**DOI:** 10.3390/jcm10132932

**Published:** 2021-06-30

**Authors:** Ainara Cano, Carlos Alcalde, Amaya Belanger-Quintana, Elvira Cañedo-Villarroya, Leticia Ceberio, Silvia Chumillas-Calzada, Patricia Correcher, María Luz Couce, Dolores García-Arenas, Igor Gómez, Tomás Hernández, Elsa Izquierdo-García, Dámaris Martínez Chicano, Montserrat Morales, Consuelo Pedrón-Giner, Estrella Petrina Jáuregui, Luis Peña-Quintana, Paula Sánchez-Pintos, Juliana Serrano-Nieto, María Unceta Suarez, Isidro Vitoria Miñana, Javier de las Heras

**Affiliations:** 1Biocruces Bizkaia Health Research Institute, 48093 Barakaldo, Spain; ainara.cano@ehu.eus; 2Paediatrics Unit, Río Hortega University Hospital, 47012 Valladolid, Spain; calcalma@saludcastillayleon.es; 3Metabolic Diseases Unit, Department of Paediatrics, Ramon y Cajal Hospital, 28034 Madrid, Spain; amaya.belanger@salud.madrid.org; 4Department of Metabolism Diseases and Nutrition, Niño Jesús University Children’s Hospital, 28009 Madrid, Spain; elvira.canedo@salud.madrid.org; 5Internal Medicine Service, Cruces University Hospital, 48903 Barakaldo, Spain; leticia.ceberiohualde@osakidetza.eus; 612 de Octubre University Hospital, CIBERER, 28041 Madrid, Spain; silvia.chumillas@salud.madrid.org (S.C.-C.); moralmon@yahoo.es (M.M.); 7Nutrition and Metabolic diseases Unit, La Fe University Hospital, 46026 Valencia, Spain; correcherpat@gva.es (P.C.); vitoriaisi@gva.es (I.V.M.); 8Unit of Diagnosis and Treatment of Congenital Metabolic Diseases, Department of Paediatrics, IDIS-Health Research Institute of Santiago de Compostela, CIBERER, MetabERN, Santiago de Compostela University Clinical Hospital, 15704 Santiago de Compostela, Spain; maria.luz.couce.pico@sergas.es (M.L.C.); Paula.Sanchez.Pintos@sergas.es (P.S.-P.); 9Department of Paediatric Gastroenterology, Hepatology and Nutrition, Sant Joan de Déu Hospital, 08950 Barcelona, Spain; dgarciaa@sjdhospitalbarcelona.org (D.G.-A.); dmartinezc@sjdhospitalbarcelona.org (D.M.C.); 10Araba University Hospital, 01009 Gasteiz, Spain; igor.gomezgarate@osakidetza.eus; 11Paediatric Service, Albacete University Hospital, 02006 Castilla-La Mancha, Spain; toherber.sescam@gmail.com; 12Pharmacy Department, Infanta Leonor University Hospital, 28031 Madrid, Spain; elsa.izquierdo@salud.madrid.org; 13Gastroenterology and Nutrition Section, Niño Jesús University Children’s Hospital, 28009 Madrid, Spain; consuelocarmen.pedron@salud.madrid.org; 14Clinical Nutrition Section, Navarra University Hospital, 31008 Pamplona, Spain; me.petrina.jauregui@navarra.es; 15Pediatric Gastroenterology, Hepatology and Nutrition Unit, Mother and Child Insular University Hospital Complex, Asociación Canaria para la Investigación Pediátrica (ACIP), CIBEROBN, University Institute for Research in Biomedical and Health Sciences, University of Las Palmas de Gran Canaria, 35016 Las Palmas de Gran Canaria, Spain; luis.pena@ulpgc.es; 16Paediatric Service, Málaga Regional University Hospital (HRU), 29010 Málaga, Spain; serranonieto@hotmail.com; 17Biochemistry Laboratory, Metabolism Area, Cruces University Hospital, 48903 Barakaldo, Spain; maria.uncetasuarez@osakidetza.eus; 18Division of Paediatric Metabolism, CIBERER, Cruces University Hospital, 48093 Barakaldo, Spain; 19Department of Paediatrics, University of the Basque Country (UPV/EHU), 48940 Leioa, Spain

**Keywords:** hereditary fructose intolerance, fructose, sucrose, sorbitol, diet, sialotransferrin profile, biomarker, aldolase B

## Abstract

Hereditary Fructose Intolerance (HFI) is an autosomal recessive inborn error of metabolism characterised by the deficiency of the hepatic enzyme aldolase B. Its treatment consists in adopting a fructose-, sucrose-, and sorbitol (FSS)-restrictive diet for life. Untreated HFI patients present an abnormal transferrin (Tf) glycosylation pattern due to the inhibition of mannose-6-phosphate isomerase by fructose-1-phosphate. Hence, elevated serum carbohydrate-deficient Tf (CDT) may allow the prompt detection of HFI. The CDT values improve when an FSS-restrictive diet is followed; however, previous data on CDT and fructose intake correlation are inconsistent. Therefore, we examined the complete serum sialoTf profile and correlated it with FSS dietary intake and with hepatic parameters in a cohort of paediatric and adult fructosemic patients. To do so, the profiles of serum sialoTf from genetically diagnosed HFI patients on an FSS-restricted diet (*n* = 37) and their age-, sex- and body mass index-paired controls (*n* = 32) were analysed by capillary zone electrophoresis. We found that in HFI patients, asialoTf correlated with dietary intake of sucrose (R = 0.575, *p* < 0.001) and FSS (R = 0.475, *p* = 0.008), and that pentasialoTf+hexasialoTf negatively correlated with dietary intake of fructose (R = −0.386, *p* = 0.024) and FSS (R = −0.400, *p* = 0.019). In addition, the tetrasialoTf/disialoTf ratio truthfully differentiated treated HFI patients from healthy controls, with an area under the ROC curve (AUROC) of 0.97, 92% sensitivity, 94% specificity and 93% accuracy.

## 1. Introduction

Hereditary fructose intolerance (HFI; OMIM 229600) is an autosomal recessive inborn error of metabolism [1]. HFI was first reported in 1956 by Chambers and Pratt [2]. It is characterized by deficiency of the enzyme fructose-1,6-bisphosphate aldolase (aldolase B; E.C. 4.1.2.13), which is predominantly expressed in the liver, kidney and small intestine [3]. Aldolase B catalyses different reactions including cleavage of fructose-1-phosphate (F1P) and reversible cleavage of fructose-1,6-bisphosphate (FBP) into glyceraldehyde phosphate and dihydroxyacetone phosphate (DHAP) [1]. Therefore, this enzyme plays a key role in the control of fructose and glucose metabolism, regulating both glycolysis and gluconeogenesis.

HFI is caused by homozygous or compound heterozygous mutations in the aldolase B gene (*ALDOB*; 612724) on chromosome 9q31 [4]. Based on the carrier frequency of the most common mutations in neonates, it has been estimated that the prevalence of HFI is around 1 in 26,000 live births in Europe [5] and 1 in 20,000 births in the US [6], yet many authors agree that it may be significantly underdiagnosed. Symptoms in HFI patients are usually initiated five or six months after birth due to the introduction of complementary feeding in the infant, who reacts with a variety of clinical signs such as failure to thrive, accompanied by vomiting, abdominal pain and acute liver failure [7]. HFI is also characterised by a set of metabolic alterations that include hypoglycaemia, metabolic acidosis, hypophosphatemia, hyperuricemia, hypermagnesemia and hyperalaninaemia after fructose loading [1]. Early diagnosis of HFI prevents the continued intake of fructose, which would otherwise lead to renal and hepatic seizures, coma and even death [1]. It allows dietary treatment, consisting mainly of a fructose-, sucrose-, and sorbitol (FSS)-restricted diet for life [1]. Recurrent symptoms make the diagnosis possible in childhood; however, many patients remain undiagnosed until adulthood [8].

Even though fructosemic patients are believed to remain quite healthy when kept on a fructose-restricted diet, compliance with this regime is often difficult, especially when taking into consideration the overload of fructose in our alimentary routines observed in recent years. There is a growing use of fructose, as it is widely used as a food additive, and small amounts of this sugar are hidden in many foods. Thus, it is not surprising that HFI patients develop previously unreported complications, such as liver steatosis [7,9] or signs of proximal tubular dysfunction [10,11].

Whereas easy detection of galactosemia, another error of carbohydrate metabolism, can be performed by means of metabolic analysis using dried blood spot (DBS) testing [12], no simple metabolic test is available for the rapid identification of HFI. The diagnosis of HFI used to be based on the clinical features of fructose intolerance, liver biopsy to assess aldolase B activity, and/or a fructose challenge test [13]. Nowadays, the diagnosis of HFI consists in identifying either biallelic pathogenic variants of *ALDOB* by molecular genetic testing or deficient hepatic aldolase B activity from a liver biopsy [13]. Despite liver biopsy being very specific, it is invasive and costly. Therefore, the high sensitivity and the non-invasive nature of *ALDOB* genetic testing make it the preferred method for HFI diagnosis [13].

In the mid-1970s, a temporary change in the transferrin (Tf) glycoform profile in the serum and cerebrospinal fluid was demonstrated to be associated with sustained heavy alcohol consumption [14]. The abnormal serum Tf pattern, initially presenting as an increased amount of Tf bands with an isoelectric point at or above 5.7 (corresponding to asialo-, monosialo-, and disialo-Tf) in isoelectric focusing (IEF), improved on abstinence, with a half-life of ~10 days [15]. This protein fraction, later named serum carbohydrate-deficient Tf (CDT), was suggested as a specific biomarker, and its measurement was proposed to identify sustained heavy alcohol consumption and monitor abstinence during treatment [16]. In vitro studies using cultured rat liver hepatocytes suggest that acetaldehyde resulting from alcohol dehydrogenase oxidation of ethanol causes the disruption of endoplasmic reticulum function and thus interferes with glycosylation [17]. At present, loss of sialylation on serum Tf is used as a screening test both for chronic alcohol consumption and for congenital disorders of glycosylation (CDG) [15,16].

Serum Tf is a glycoprotein that transports iron (Fe^3+^), which is mainly synthesised and metabolised in liver hepatocytes. Two complex chains of oligosaccharides, which vary in their degree of ramification, form the glycolic portion of the glycoprotein; each of them can have two or three external chains or anthems, with a sialic acid residue in the terminal position (Figure 1). Then, Tf has different isoforms depending on the number of sialic acid residues present on its oligosaccharide chain: asialo-, monosialo-, disialo-, trisialo-, pentasialo- and hexasialo-Tf. As stated by Helander et al. [15], Tf shows natural microheterogeneity, owing to variations in iron load, amino acid sequence (i.e., genetic variants) and the structure of the two N-linked oligosaccharides (N-glycans). In human serum, tetrasialoTf is usually the most abundant glycoform (~80%), followed by pentasialo- (~14%), trisialo- (~4%), disialo- (~1%) and hexasialoTf (~1%) [18,19].

Previous studies have shown that untreated fructosemic patients have an abnormal Tf glycosylation pattern as a consequence of F1P-mediated competitive inhibition of mannose-6-phosphate isomerase (MPI) [20]. In particular, patients with HFI present a Tf isoelectric focussing (IEF) type Ib pattern, in which defects in the synthesis and assembly of the glycans occur, indicating that HFI is a secondary CDG syndrome [21]. Therefore, as glycosylation of Tf is a measure of intrahepatic F1P concentrations, elevated serum CDT values allow the quick detection of HFI [22]. This biomarker is valuable for detecting the persistence of some abnormalities caused by trace amounts of fructose ingestion and/or non-adherence to the prescribed diet in HFI patients. Moreover, CDT values improve after 2–4 weeks when a fructose-restricted diet is taken, and then, CDT measurements are recommended for HFI diagnosis and treatment monitoring [22].

However, there is no agreement among authors regarding the correlation of the sialoTf profile with dietary fructose consumption, nor about which isoform of Tf is more valuable for monitoring fructosemic patients [21,22,23,24,25]. Thus, the objective of the present study was to determine the usefulness of the entire sialoTf profile for monitoring an FSS-restricted diet in HFI patients by assessing the correlation between the different sialoTf isoforms and dietary fructose, sucrose, and sorbitol intake.

## 2. Experimental Section

### 2.1. Participants

A cross-sectional study was conducted from October 2019 to November 2020. The study population comprised 37 genetically diagnosed HFI patients from 31 unrelated families. The recruited fructosemic patients had been on dietary treatment with fructose, sucrose and sorbitol exclusion for at least two years. Their age-, sex-, and BMI-matched healthy controls (*n* = 32) were also studied.

Eleven Spanish hospitals participated in the study: Cruces University Hospital, Basque Country [host]; Araba University Hospital, Basque Country; Navarra University Hospital; Navarra, 12 de Octubre University Hospital, Madrid; Niño Jesús University Children’s Hospital, Madrid; Ramón y Cajal University Hospital, Madrid; La Fe University Hospital, Valencian Community; Málaga Regional University Hospital, Andalusia; Santiago de Compostela University Clinical Hospital, Galicia; Río Hortega University Hospital, Castile and Leon; Mother and Child Insular University Hospital complex, Canary Islands.

The study protocol was performed according to the ethical guidelines of the revised 1975 Declaration of Helsinki [26] and approved by the Research Ethics Committee of the Basque Country (CEIm-E), ethic approval code: PI2019072. Written informed consent was obtained from parents or legal guardians of children (below 18 years of age) and adult study participants.

### 2.2. Study Visits

Study visits were performed at the Cruces University Hospital, Spain. Blood samples were collected at the same time, at 8:30 a.m., after at least 8 h of fasting. Patients were weighed, and their height and abdominal perimeters were measured. Standing height was measured with a wall-mounted stadiometer, and patients were weighed barefoot, to the nearest 100 g, with digital scales.

### 2.3. Genotype Analysis

HFI patients were genetically diagnosed in their hospitals of origin by Sanger sequencing of the *ALDOB* gene [27] or by NGS gene panel screening.

### 2.4. Biochemical Analyses

Concentrations of iron, Tf and ferritin in the plasma were determined by routine clinical techniques.

### 2.5. Transferrin Quantification

The Tf profile was determined in serum by capillary zone electrophoresis (CZE) using a commercially available system (Sebia, Capillarys 2TM, France) in 37 HFI patients and in 32 healthy volunteers as described before [18]. Briefly, the Tf isoforms were separated into five fractions according to their degree of sialylation, i.e., asialoTf (non-sialylated form), disialo-, trisialo-, tetrasialo- and the sum of pentasialo- and hexasialo-Tf, and expressed as a percentage of the total sialoTf. Data analysis was performed with the software package Phoresis 8.6.3 (Sebia, France). The sum of the asialoTf and disialoTf fractions was expressed as a CDT percentage. Additionally, the tetrasialoTf-to-disialoTf ratio was calculated.

### 2.6. Assessment of Diet

Dietary information was collected in a self-administered nutritional record of dietary intake on three days (two during the week and one on the weekend) in 34 of the 37 HFI patients, with a special emphasis on determining fructose, sucrose and sorbitol dietary intake. Once completed, the different carbohydrate composition, expressed in mg per day, was calculated using the Nutritional Calculation DIAL Program (Version 3.10.5.0), of ALCEGENI INERIA [28]. In addition, to complete the information of fructose and sorbitol dietary intake, the following databases were consulted: Australian Food Composition Database—Release 1.0 [29], Danish Frida Food Database [30], and German Food Composition and Nutrition Tables [31]. Data obtained on fermentable mono-, di-, oligosaccharides and polyols (FODMAP) were also used [32].

### 2.7. Statistical Analysis

Continuous variables were represented as mean ± standard deviation and range. Fructosemic patients on an FSS-restrictive diet (*n* = 37) were compared with their age-, sex-, and BMI-marched controls (*n* = 32) by unpaired Student’s *t*-test. The Mann–Whitney U test was used to compare continuous variables between HFI patients with and without asialoTf expression. the Wallis test was used to compare continuous variables among the FFS-intake tertiles. *p*-values were based on two-tailed comparisons as appropriate, and those less than 0.05 were considered to indicate a statistically significant difference.

Pearson correlation was used to assess bivariate relationships between variables. The diagnostic accuracy of the model that differentiates between HFI patients and non-HFI patients was assessed by the area under the ROC curve (AUC) by exporting the obtained data to the MetaboAnalyst software package (version 5.0). The optimal score cut-off value for the estimation group was selected based on sensitivity and specificity, and positive and negative likelihood ratios as that at which the sum of sensitivity and specificity was maximum.

Statistical analyses were performed using SPSS software, version 23 for Windows (IBM, Chicago, IL), R Software (R version 3.2.0; R Development Core Team, 2010; http://cran.r-project.org) and Microsoft Office Excel (v19.0).

## 3. Results

### 3.1. Study Visits

The study population comprised 37 genetically diagnosed HFI patients from 31 unrelated families and their 32 healthy controls (Table 1). The HFI patients were 22 females and 15 males; 23 patients were below 18 years of age, and 14 were adults. The controls were 19 females and 13 males, and of these volunteers, 20 were below 18 years of age, and 12 were adults.

The majority of the HFI patients enrolled were diagnosed during their childhood, i.e., at 3.0 ± 2.6 years, except four patients, who were diagnosed at 34, 43, 48,and 63 years of age. There were no differences in weight, height, BMI or waist circumference between the HFI patients and their respective controls (Table 1).

### 3.2. Biochemical Analyses

The relative percentages of all sialoTf isoforms analysed were altered in HFI patients when compared with their controls, as shown in Table 1 and in Figure 2. The concentration of CDT, disialo-, trisialo- and the sum of pentasialo- and hexasialo-Tf in the HFI patients was higher by 3.7-fold, 3.5-fold, 28%,and 23%, respectively, when compared with the controls. In contrast, tetrasialoTf concentration was lower by 6% in the HFI patients in comparison with their healthy controls. AsialoTf was present in 6 out of the 37 HFI patients and was not detected in any control participant.

### 3.3. Serum Transferrin Isoforms, Hepatic Parameters and Fructose Consumption in Patients with Fructosemia

Table 2 shows Tf isoforms, hepatic parameters and FSS consumption in HFI patients, all together and divided by FSS-intake tertiles, and shows that there were no significant differences in sialoTf or the studied hepatic parameters among the three tertiles of FSS-intake in HFI participants.

There were six fructosemic participants with asialoTf expression. There were no significant differences in the studied hepatic parameters or in the FSS-intake between HFI patients who expressed asialoTf and those who did not express it (Table 3).

### 3.4. Correlations between Transferrin Isoforms, Hepatic Parameters and Fructose Consumption in Patients with Fructosemia

The correlations between the dietary intake of fructose, sucrose, sorbitol and their sum (FSS) with the entire sialoTf profile were assessed in HFI patients (Table 4). The highest correlation values were found for the asialoTf percentage, which correlated with dietary intake of sucrose (R = 0.575, *p* < 0.001) and FSS (R = 0.475, *p* = 0.008). The pentasialoTf+hexasialoTf fraction correlated negatively with fructose (R = −0.386, *p* = 0.024) and with FSS intake (R = −0.400, *p* = 0.019), and tetrasialoTf correlated with dietary intake of sorbitol (R = 0.361, *p* = 0.036).

No significant correlations were found for CDT, disialoTf, trisialoTf or for the ratio of tetrasialoTf to disialoTf with dietary intake in HFI patients.

The studied hepatic parameters did not significantly correlate with the sialoTf isoforms (Table 4).

### 3.5. Diagnostic Accuracy of Serum Transferrin Isoforms in HFI

The highest predictive values for sialoTf were displayed by the ratio of tetrasialoTf to disialoTf; this quotient had an AUROC of 0.97, 92% sensitivity, 94% specificity, 14.7 positive likelihood ratio, 0.09 negative likelihood ratio and 93% accuracy with a cut-off point of 93 (Figure 3). 

CDT yielded an AUROC of 0.96, 95% sensitivity, 91% specificity, 10.1 positive likelihood ratio, 0.06 negative likelihood ratio and 83% accuracy with a cut-off point of 0.85 (Figure 3).

## 4. Discussion

To date, all the studies carried out with regard to sialoTf in HFI have focused on the determination of the corrective effects of a fructose-restrictive diet on the profile of sialoTf [21,22,23,24,25]. In addition, published reports have been CDT-centred [21,22,24] and sometimes, limited to single cases [21,23]. In an attempt to clarify the existing inconsistency concerning sialoTf found in the literature, the complete serum sialoTf profile of genetically diagnosed HFI patients under a FSS-restrictive diet was examined, compared to that of their age-, sex- and body mass index-matched healthy controls and correlated with the dietary intake of fructose, sucrose and sorbitol and with hepatic parameters.

Previous studies have shown that untreated HFI patients have an abnormal Tf glycosylation pattern [21,22,23,24,25]. While glycosylation of Tf is a measure of intrahepatic F1P concentrations, elevated serum CDT values have been proposed as a way of allowing the prompt detection of fructosemia and treatment monitoring [22,25]. In this study, we observed that not only the percentage of CDT, but also the complete sialoTf profile was profoundly altered in 37 HFI patients that followed an FSS-restrictive diet when compared with their controls. Interestingly, we found that the percentages of asialoTf, disialoTf, trisialoTf and pentasialoTf+hexasialoTf were higher and that the tetrasialoTf fraction was very significantly lower in HFI patients when compared with their controls. Thus, the percentage of CDT, considered as the sum of asialoTf and disialoTf, was also higher in fructosemic patients when compared with their healthy controls. Similarly, in 1996, Adamowicz and Pronicka [23] observed that the asialoTf and disialoTf isoforms had an upward trend and that tetrasialoTf had a downward trend in an HFI patient treated with a fructose-restricted diet when compared to a non-HFI control.

Here, we report that the ratio of tetrasialoTf to disialoTf could be an accurate and precise biomarker for distinguishing between treated HFI patients and non-HFI subjects, considering both children and adults, with an AUROC of 0.97, 92% sensitivity, 94% specificity and 93% accuracy. The ratio between tetrasialoTf and disialoTf turns out to be a better marker than the traditionally analysed CDT parameter, which had lower accuracy (83%), with an AUROC of 0.96, 95% sensitivity and 91% specificity. Certainly, it would be relevant to consider the use of this ratio for HFI detection in non-diagnosed patients, as many of these patients follow an intuitive diet that is analogous to that of diagnosed patients [33]. Furthermore, given that in human serum, tetrasialoTf is usually the most abundant glycoform (~80%) and that disialoTf concentration is ~1% [19], the ratio of tetrasialoTf to disialoTf proposed here as a biomarker represents the highest percentage of the total sialoTf fractions.

In our study, there was no significant correlation between the sialoTf isoforms and the studied hepatic parameters. Also, there were no significant associations between the different HFI genetic variants and the sialoTf isoforms (data not shown). Regarding sugar dietary intake, among the different sialoTf isoforms, asialoTf presented the highest correlation coefficients with FSS dietary intake, with a statistically significant correlation with the dietary intake of sucrose (R = 0.575) and FSS (R = 0.475). The pentasialoTf+hexasialoTf fraction correlated negatively with dietary intake of fructose and FSS in 34 fructosemic patients, and the tetrasialoTf percentage correlated with their dietary sorbitol intake. Unlike us, Pronicka et al. [22] did not find a clear statistical correlation between serum CDT values and fructose consumption in eight HFI patients. More recently, in 2019, Di Dato et al. [25] described that fructose intake was significantly correlated with disialoTf and with the ratio of serum tetrasialoTf to disialoTf in 38 HFI patients. The sialoTf fractions reported by this group were different from those that we found significantly correlated with the dietary intake of fructose in our study, i.e., pentasialoTf + hexasialoTf. Moreover, this was a prospective study of HFI patients before and after dietary treatment and, contrary to the rest of the studies and to ours, where we analysed serum samples, they used dried blood spots for sialoTf determination. In addition, since the sialoTf profile has not been correlated with fructose precursors before, i.e., sucrose and sorbitol, the results reported in our study could shed light on treatment monitoring in HFI patients.

The CDT fraction is defined as the sum of the glycoforms with isoelectric points at or above pH 5.7 after complete iron saturation, corresponding to asialo-, monosialo- and disialo-Tf [16]. However, depending on the procedure used to determine the sialoTf profile, results can lack uniformity and hamper comparison [15]. Profiling of sialoTf isoforms can be performed by electrophoretic and chromatographic methods. The earliest and most frequently used method to define glycosylation defects is isoelectric focusing of serum sialoTf [34]. Ever since this technique was used by the groups of Pronicka [22] and Di Dato [25], one can assume that the calculated CDT percentages include asialo-, monosialo- and disialo-Tf. Yet, in general, the percentages of the monosialoTf and asialoTf fractions tend to be too low, and in addition, CDT normally essentially represents the disialoTf fraction. In 2018, Kinma et al. [18] demonstrated that capillary zone electrophoresis (CZE) was a fast, reliable method for screening N-glycosylation defects. This technique was used in the present study, in which asialoTf was detected in 6 patients out of 37 and was absent in the 32 control volunteers. Although dietary FSS intake and asialoTf were significantly correlated in our study, there were no significant differences in the FFS intake between HFI patients who expressed asialoTf (*n* = 6) and those who did not express it (*n* = 31), probably due to the small number of participants who expressed asialoTf. Also, there were no differences in the studied hepatic parameters between these two groups.

Recurrent symptoms make the diagnosis of this dysfunction possible in childhood; however, many HFI patients remain undiagnosed until adulthood [8]. The vast majority of the fructosemic patients included in this study were diagnosed during their childhood, i.e., at 3.0 ± 2.6 years, except four patients, who were diagnosed at 34, 43, 48 and at 63 years of age. These patients did not show different sialoTf levels or carbohydrate consumption with respect to those diagnosed when they were infants (data not shown).

A perceived weakness of this study is that, as occurs with rare diseases, the statistical power would improve with a larger cohort of participants. However, we were able to enrol a fairly good number of fructosemic patients and healthy controls, 37 and 32, respectively. In addition, although the same researcher instructed all participants, the nutritional records of dietary intake could be subjective and susceptible to underreporting.

In summary, the results shown here evidence that the entire sialoTf profile should be considered for the diagnosis and follow-up of HFI patients, even on an FSS-restricted diet. To our knowledge, this is the first time that the ratio of tetrasialoTf to disialoTf has been demonstrated to be an accurate biomarker for identifying HFI patients. Moreover, the asialoTf and pentasialoTf + hexasialoTf isoforms can be considered valuable indicators for treatment monitoring of FSS dietary intake in patients with fructosemia.

## Figures and Tables

**Figure 1 jcm-10-02932-f001:**
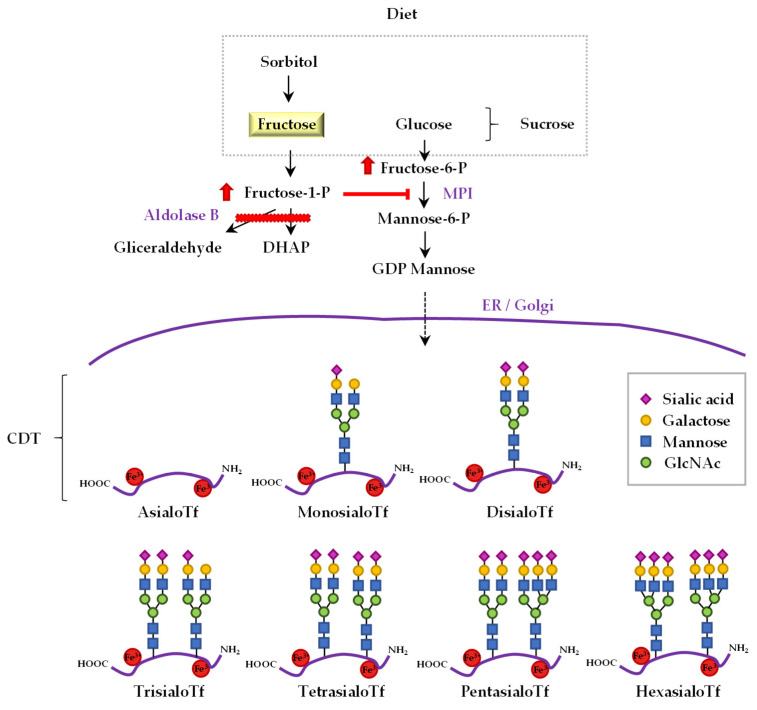
The link between fructose metabolism and the profile of sialotransferrins in hereditary fructose intolerance. Dietary sorbitol and sucrose, i.e., a disaccharide formed by glucose and fructose, are precursors of fructose. The catabolic pathway of dietary fructose, sucrose and sorbitol is altered in patients with hereditary fructose intolerance (HFI). In aldolase B deficiency, the catabolism of fructose-1-P (F1P) is impaired (red bar), and this molecule is accumulated largely in the liver of HFI patients. Consequently, HFI patients have an abnormal transferrin (Tf) glycosylation pattern because of F1P-mediated competitive inhibition of mannose-6-phosphate isomerase (MPI). Tf exhibits different isoforms depending on the number of sialic acid residues present on its oligosaccharide chain; asialo-, monosialo-, disialo-, trisialo-, pentasialo-, and hexasialo-Tf. The sum of asialo-, monosialo-, and disialo-Tf is called carbohydrate-deficient Tf (CDT).

**Figure 2 jcm-10-02932-f002:**
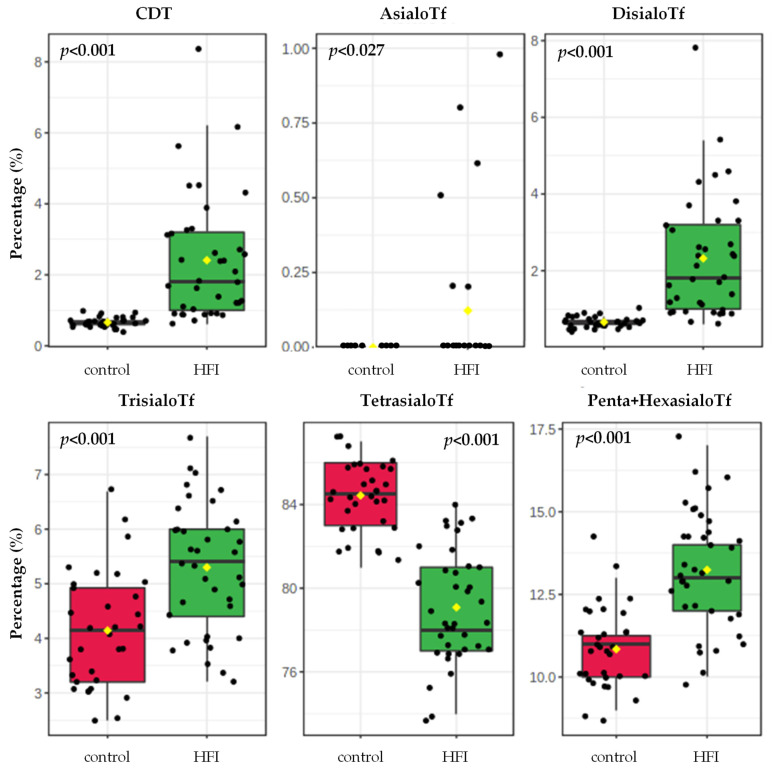
Sialotransferrin (sialoTf) isoform percentages in HFI patients (*n* = 37) and in their respective healthy controls (*n* = 32). Differences in continuous variables between HFI and their controls were calculated by using the Student’s *t*-test.

**Figure 3 jcm-10-02932-f003:**
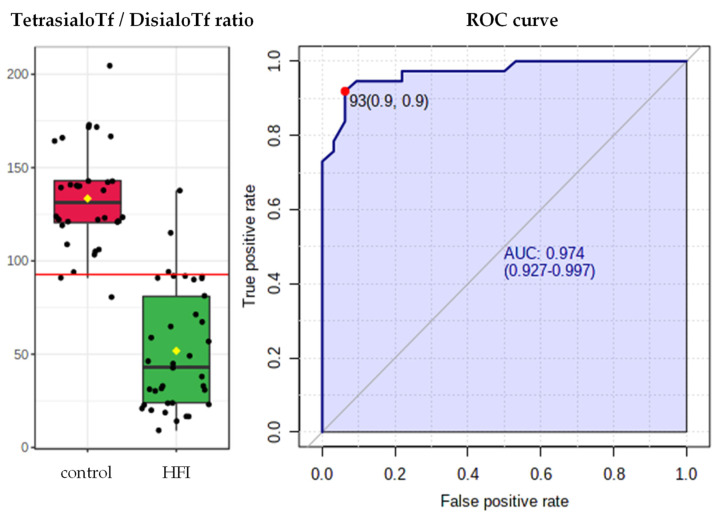
TetrasialoTf-to-disialoTf ratio and ROC curve in HFI patients (*n* = 37) and in their respective healthy controls (*n* = 32). AUROC = 0.97, 92% sensitivity, 94% specificity, 14.7 positive likelihood ratio, 0.09 negative likelihood ratio and 93% accuracy with a cut-off point of 93.

**Table 1 jcm-10-02932-t001:** Principal features in HFI patients and in their healthy controls. Continuous variables are represented as mean ± standard deviation and range. Differences in continuous variables between HFI patients and controls were calculated by using the Student’s *t*-test. Transferrin (Tf).

	Controls	HFI Patients	*p*-Value
*n*	32	37	
Women/men, *n*/*n*	19/13	22/15	0.594
Age, year	21.5 ± 15.1 (2.1–61.3)	19.7 ± 14.8 (3.7–63.4)	0.629
Weight, kg	50.5 ± 15.3 (11.9–71.0)	45.3 ± 16.4 (17–73)	0.178
Height, cm	155.5 ± 19.5 (85.5–181.0)	151.0 ± 21.1(104.2–190.7)	0.368
BMI, kg/m^2^	20.2 ± 3.2 (14.6–27.1)	19.0 ± 3.1 (14.0–27.4)	0.107
Waist circumference, cm	68.3 ± 9.4 (48–88)	67.5 ± 9.7 (51–87)	0.736
CDT, % (asialoTf + disialoTf)	0.7 ± 0.1 (0.4–1.0)	2.4 ± 1.8 (0.6–8.4) 9	**<0.001**
AsialoTf, %	0.0 ± 0.0 (0.0–0.0)	0.10 ± 0.3 (0.0–0.1)	**0.027**
DisialoTf, %	0.7 ± 0.1 (0.4–1.0)	2.3 ± 1.6 (0.6–7.8)	**<0.001**
TrisialoTf, %	4.4 ± 1.7 (2.5–6.7)	5.3 ± 1.2 (3.2–7.7)	**<0.001**
TetrasialoTf, %	84.4 ± 1.6 (81–87)	79.0 ± 2.7 (74–84)	**<0.001**
Penta- and hexa-sialoTf, %	10.8 ± 1.1 (9.0–14.1)	13.3 ± 1.7 (10.3–17.1)	**<0.001**
Tetra-/di-sialoTf	133.4 ± 27.7 (81.3–204.8)	51.8 ± 32.9 (9.4–137.5)	**<0.001**
Serum iron (mg/dL)	98.7 ± 39.7 (34–205)	85.3 ± 30.6 (53–216)	0.133
Tf (mg/dL)	270.9 ± 37.8 (210–397)	288.1 ± 36.5 (222–378)	0.064
Ferritin (mg/dL)	50.4 ± 37.9 (10–176)	73.2 ± 73.4 (4–284)	0.124

Significant *p*-values are marked in bold.

**Table 2 jcm-10-02932-t002:** Dietary intake of fructose, sucrose, sorbitol and their sum (FSS), sialoTf percentages and liver parameters in HFI patients, all together (*n* = 34) and divided into FSS-intake tertiles: 1st (*n* = 12), 2nd (*n* = 11) and 3rd (*n* = 11). Continuous variables are represented as mean ± standard deviation and range. Differences in continuous variables among the three tertiles were calculated using the Kruskal–Wallis test.

	HFI Patients all Together	FSS-Intake 1st Tertile	FSS-Intake 2nd Tertile	FSS-Intake 3rd Tertile	*p*
*n*	34	12	11	11	
Fructose intake (mg/day)	322 ± 304 (0–1323)	70 ± 78 (0–220)	296 ± 161 (98–604)	621 ± 311 (238–1323)	**<0.001**
Sucrose intake (mg/day)	503 ± 651 (15–3747)	131 ± 99 (15–308)	378 ± 70 (255–530)	1032 ± 944 (344–3747)	**<0.001**
Sorbitol intake (mg/day)	50 ± 93 (0–443)	11 ± 20 (0–57)	32 ± 40 (0–90)	11 ± 142 (0–443)	**0.049**
FSS intake (mg/day)	1764 ± 938 (15–4366)	212 ± 157 (15–446)	707 ± 193 (457–936)	1764 ± 937 (1020–4366)	**<0.001**
AsialoTf, %	0.08 ± 0.25 (0–0.8)	0.08 ± 0.21 (0.0–0.6)	0.09 ± 0.22 (0.0–0.8)	0.14 ± 0.32 (0.00–1.00)	0.934
DisialoTf, %	2.3 ± 1.6 (0.6–7.8)	2.2 ± 2 (0.6–7.8)	2.4 ± 1.5 (0.9–5.4)	2.3 ± 1.3 (0.9–4.6)	0.612
TrisialoTf, %	5.3 ± 1.2 (3.2–7.7)	5.1 ± 1.1 (3.4–6.8)	5.2 ± 1.2 (3.2–7.1)	5.7 ± 1.1 (4.0–7.7)	0.395
TetrasialoTf, %	79.0 ± 2.7 (73.7–84.3)	78.6 ± 2.9 (73.7–82.5)	79.6 ± 2.9 (74.4–84.3)	78.7 ± 2.1 (76.7–82.7)	0.674
Penta- and hexa-sialoTf, %	13.3 ± 1.7 (10.3–17.1)	14.1 ± 1.8 (10.8–17.1)	12.7 ± 1.6 (10.3–15.9)	13.1 ± 1.5 (10.37–14.6)	0.262
Tetra-/di-sialoTf	51.8 ± 32.9 (9.4–137.5)	62.2 ± 40.8 (9.4–137.5)	47.3 ± 29.1 (13.8–93.7)	46.1 ± 27.9 (16.7–91.9)	0.758
GGT (U/L)	15.3 ± 5.9 (9–36)	17.4 ± 8.2 (9–36)	14.4 ± 4.9 (9–27)	14.2 ± 3.5 (10–21)	0.221
GOT, AST (U/L)	26.5 ± 9.5 (16–66)	45.8 ± 30.4 (19–45)	28.5 ± 4.8 (19–33)	27.9 ± 4.7 (16–66)	0.322
GPT, ALT (U/L)	30.4 ± 31.4 (9–198)	44.3 ± 50.7 (16–198)	22.8 ± 9.5 (10–47)	24.7 ± 15.7 (9–68)	0.234
INR	1.00 ± 0.08 (0.9–1.1)	0.96 ± 0.08 (0.9–1.1)	1.00 ± 0.67 (0.9–1.1)	1.00 ± 0.77 (0.9–1.1)	0.151

Significant *p*-values are marked in bold.

**Table 3 jcm-10-02932-t003:** Dietary intake of fructose, sucrose, sorbitol, their sum (FSS), liver parameters and sialoTf isoform percentages in HFI patients with (*n* = 6) or without (*n* = 31) the asialoTf isoform. Continuous variables are represented as mean ± standard deviation and range. Differences in continuous variables between HFI patients with or without asialoTf isoform were calculated using the Mann–Whitney U test. Transferrin (Tf).

	No AsialoTf Expression	AsialoTf Expression	*p*-Value
Fructose intake (mg/day)	354 ± 321 (0–1323)	434 ± 327 (220–620)	0.254
Sucrose intake (mg/day)	434 ± 327 (15–1334)	1096 ± 1496 (117–3747)	0.232
Sorbitol intake (mg/day)	64 ± 104 (0–443)	19 ± 41 (0–93)	0.192
FSS intake (mg/day)	868 ± 596 (15–2119)	1535 ± 1618 (337–4366)	0.487
GGT (U/L)	15.5 ± 6.1 (9–36)	14.7 ± 5.2 (9–21)	1
GOT, AST (U/L)	27.2 ± 10.3 (13–66)	23 ± 2.8 (19–27)	0.606
GPT, ALT (U/L)	31.5 ± 34.2 (9–198)	24.7 ± 6.2 (19–35)	0.223
INR	1.0 ± 0.1 (0.9–1.1)	1.0 ± 0.1 (1.0–1.1)	0.295
CDT, % (asialoTf+disialoTf)	1.8 ± 1.0 (0.6–4.5)	5.5 ± 1.7 (3.9–8.4)	**<0.001**
AsialoTf, %	0	0.6 ± 0.3 (0.2–1.0)	**<0.001**
DisialoTf, %	1.8 ± 1.0 (0.6–4.5)	4.9 ± 1.5 (3.7–7.8)	**<0.001**
TrisialoTf, %	5.3 ± 1.2 (3.2–7.7)	5.6 ± 1.2 (4.0–7.1)	0.575
TetrasialoTf, %	79.5 ± 2.5 (76.5–84.3)	76.5 ± 2.2 (73.5–79.8)	**0.005**
Penta- and hexa-sialoTf, %	58.6 ± 31.7 (10.3–17.1)	16.6 ± 4.3 (10.3–14.4)	0.427
Tetra-/di-sialoTf	13.4 ± 1.7 (17.0–137.5)	12.5 ± 1.6 (9.4–20.9)	**<0.001**

Significant *p*-values are marked in bold.

**Table 4 jcm-10-02932-t004:** Correlation analysis of sialotransferrin (Tf) fraction levels with dietary intake of fructose, sucrose, sorbitol and their sum (FSS) and with liver parameters in HFI patients. Pearson correlation analysis was used to assess bivariate relationships between variables. Significant *p*-values are marked in bold.

Dietary Intake		CDT, %	A-Tf, %	Di-Tf, %	Tri-Tf, %	Tetra-Tf, %	Penta- and Hexa-Tf, %	Tetra-/di-Tf Ratio
**Fructose**	**R**	0.063	0.107	0.046	0.244	0.092	−0.386	−0.085
	***p***	0.725	0.573	0.794	0.169	0.606	**0.024**	0.631
**Sucrose**	**R**	0.242	0.575	0.18	0.238	−0.073	−0.303	−0.187
	***p***	0.168	**<0.001**	0.309	0.175	0.68	0.08	0.29
**Sorbitol**	**R**	−0.197	−0.175	−0.197	−0.181	0.361	−0.253	0.209
	***p***	0.264	0.354	0.264	0.305	**0.036**	0.15	0.236
**FSS**	**R**	0.187	0.475	0.133	0.25	0.016	−0.4	−0.151
	***p***	0.29	**0.008**	0.453	0.154	0.929	**0.019**	0.392
**Liver Parameters**								
**GGT**	**R**	0.061	0.006	0.077	0.049	−0.12	0.091	−0.107
	***p***	0.719	0.974	0.653	0.774	0.48	0.592	0.528
**GOT (AST)**	**R**	−0.213	−0.135	−0.212	−0.004	0.184	−0.059	0.184
	***p***	0.212	0.453	0.214	0.982	0.282	0.733	0.281
**GPT (ALT)**	**R**	0.044	−0.01	0.061	0.076	−0.185	0.193	−0.092
	***p***	0.795	0.955	0.72	0.656	0.272	0.253	0.587
**INR**	**R**	−0.026	0.082	−0.033	−0.064	−0.055	0.162	0.103
	***p***	0.882	0.662	0.849	0.717	0.752	0.351	0.555

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
