# Peer review of "Transferrin Isoforms, Old but New Biomarkers in Hereditary Fructose Intolerance"

_jcm, 2021, doi:10.3390/jcm10132932_

Round 1

Reviewer 1 Report

The Authors provided an interesting study and useful in a clinical practice on serum Tf isoforms assessment in patients with HFI.

However, I have some issues to be revised/answered.

  1. Principal features in HFI patients and their healthy controls. The authors presented only some laboratory markers. What about liver function tests, including ALT, INR? What about liver ultrasound? There is known that HFI could be a cause of liver steatosis, increase of ALT and INR (hepatocytes damage). It will be essential information about the correlation of these parameters with serum Tf isoforms.
  2. Asialo-Tf was detected in 6/37 HFI patients. Please, provide more information about this sub-group of patients (especially regarding liver involvement). 

Author Response

We would like to thank the reviewers for their review of our manuscript and their insightful comments. We believe that our manuscript is much improved with the incorporation of the suggested comments.

Our point-by-point response to each reviewer’s comment is detailed in the enclosed document.

Reviewer 2 Report

This is a study on using carbohydrate deficient transferrin that measured by capillary zone electrophoresis, as biomarkers, to diagnose HFI and to monitor the FSS intake in 37 affected patients. It is well executed by the authors. I have  a few suggestion:

  1. The discussion section is too long and needs to be more concise
  2. Figure 2, the data from CDT, asialo and disialo, appears to have a subgroup present.  In particular, for asialo Tf,  only 6 data points are above zero. Authors stated that all 6 patients have higher FSS intake, how about the other 31 patients? None of the 31 HFI patients has high FSS intake? Is it possible other factors may also be in play, such as the HFI genotype? Since authors state in the manuscript that these patients were diagnosed by molecular testing, a master table that shows all the available mutation data,  FSS intake, and clinical outcome should be included as a supplemental table.
  3. Representative electrophoresis chromatograms need to be presented for HFI with high, average and low FSS intake, in that way, we can evaluate how the Tf variants are differentiated from different CDT glycoforms. 

Author Response

(The authors gave the same response as above.)

Round 2

Reviewer 1 Report

The Authors responded to all my questions/issues.

I appreciate Their work.

I recommend to publish the manuscript in its current form.